# Bone morphogenic protein signalling suppresses differentiation of pluripotent cells by maintaining expression of E-Cadherin

Mattias Malaguti, Paul A Nistor, Guillaume Blin, Amy Pegg, Xinzhi Zhou, Sally Lowell*

MRC Centre for Regenerative Medicine, Institute for Stem Cell Research, School of Biological Sciences, University of Edinburgh, Edinburgh, United Kingdom

**Abstract** Bone morphogenic protein (BMP) signalling contributes towards maintenance of pluripotency and favours mesodermal over neural fates upon differentiation, but the mechanisms by which BMP controls differentiation are not well understood. We report that BMP regulates differentiation by blocking downregulation of Cdh1, an event that accompanies the earliest stages of neural and mesodermal differentiation. We find that loss of Cdh1 is a limiting requirement for differentiation of pluripotent cells, and that experimental suppression of Cdh1 activity rescues the BMP-imposed block to differentiation. We further show that BMP acts prior to and independently of Cdh1 to prime pluripotent cells for mesoderm differentiation, thus helping to reinforce the block to neural differentiation. We conclude that differentiation depends not only on exposure to appropriate extrinsic cues but also on morphogenetic events that control receptivity to those differentiation cues, and we explain how a key pluripotency signal, BMP, feeds into this control mechanism.

*For correspondence:
sally.lowell@ed.ac.uk

**Competing interests:** The authors declare that no competing interests exist.

## Introduction

The events that drive the transition from pluripotency to lineage commitment remain poorly understood. It has been known for many years that BMP signalling protects pluripotency and suppresses neural specification (*Ying et al., 2003a*; *Stern, 2006*; *Di-Gregorio et al., 2007*), but the underlying mechanisms remain unclear. BMP signalling is also required for the formation of mesoderm *in vivo* (*Mishina et al., 1995*; *Winnier et al., 1995*; *Lawson et al., 1999*; *Beppu et al., 2000*; *Davis et al., 2004*), and BMP is commonly used to induce mesoderm from embryonic stem (ES) cells (*Murry and Keller, 2008*). However, it is not clear how the effects of BMP on mesoderm differentiation relate to its pro-pluripotency and anti-neural effects: are these separable independent events or do they represent the outcomes of one common mechanism? This question underlines our poor understanding of the mechanisms by which BMP influences neural and mesodermal specification.

BMP acts through transcriptional upregulation of Inhibitor of Differentiation (Id) factors (*Ying et al., 2003a*; *Zhang et al., 2010*) in order to prevent neural differentiation. Id factors generally act as dominant negative inhibitors of the bHLH family of transcription factors (*Norton, 2000*), but the mechanism by which Id proteins block neural induction is not known. Furthermore, it is not clear to what extent the pro-mesoderm effect of BMP within the epiblast is mediated by Id or by other BMP target genes: redundancy between the four Id family members may mask gastrulation phenotypes in Id mutants.

We set out to examine more closely the effects of BMP and Id1 on neural and mesoderm differentiation by taking advantage of an ES cell differentiation system, which allows differentiation to be carefully monitored in a well-defined environment (*Ying and Smith, 2003*), and by using a reporter strategy to ask which cells usually express Id1 during early development.

**eLife digest** The human body is made up of about 200 different types of cell, all of which are descended from a single fertilised egg. As an embryo develops, its cells divide and specialise into distinct lineages. Cells in each lineage go on to form a restricted number of cell types that are required to make a specific tissue. As such, during early development, cells switch from being 'pluripotent', with the potential to become the many different cell types, to committing to one particular cell lineage.

Controlling this process involves a huge number of signalling proteins and pathways. One such protein is bone morphogenetic protein, or BMP for short, which has a number of different roles in embryo development: for example, it stops pluripotent cells turning into nerve tissue, and it also encourages embryonic stem cells to contribute to the 'mesoderm' of the early embryo (which goes on to form the muscles, connective tissues and some blood cells). How these two actions are linked, and whether they depend on similar signalling pathways, was unknown.

BMP is also known to trigger the production of proteins known as 'Id factors'—which stands for 'inhibitor of differentiation'. Now, Malaguti et al. have investigated the roles of BMP and Id factors in controlling mouse embryo development and found, somewhat surprisingly, that these proteins needed help from a third protein to stop pluripotent cells turning into nerve tissue. This third protein, which is called E-Cadherin, normally helps cells to adhere to other cells. Malaguti et al. showed that losing this protein encourages cells to become either nerve or mesoderm tissues, and that a drop in E-Cadherin levels must occur before nerve tissue can form.

Malaguti et al. also showed that encouraging cells to become part of the mesoderm requires BMP to activate another pathway, which does not require E-Cadherin. The two effects of BMP can be uncoupled by adjusting the levels of this protein. At low concentrations, BMP can keep cells pluripotent, but it cannot encourage cells to commit to a mesoderm fate. At higher doses, however, BMP 'primes' cells to respond to the signals that trigger their development into mesoderm tissue.

The findings of Malaguti et al. suggest that manipulating both E-Cadherin and BMP signalling could improve our ability to generate useful cell types, such as neurons, from stem cells grown in laboratory cultures.

We find an unanticipated ability of BMP/Id to block differentiation by maintaining the expression of the cell adhesion molecule E-Cadherin (Cdh1). We find that loss of Cdh1 is tightly associated with neural as well as mesodermal differentiation and that this change in Cdh1 is a limiting requirement for neural differentiation. A number of recent reports (*Chou et al., 2008*; *Soncin et al., 2009*; *Li et al., 2010*; *Redmer et al., 2011*; *del Valle et al., 2013*; *Faunes et al., 2013*) suggest that Cdh1 helps protect pluripotency. Despite this emerging appreciation of Cdh1 as a regulator of the pluripotent state, the upstream regulators of Cdh1 in pluripotent cells have not been reported. BMP favours mesenchymal to epithelial transitions in other contexts (*Kondo et al., 2004*; *Samavarchi-Tehrani et al., 2010*), but its ability to control Cdh1 activity during early fate specification has not previously been appreciated.

We also find that BMP acts through Id1 to impose a proximal posterior identity on epiblast cells, priming them for mesodermal fates whilst transiently restraining them from overt mesoderm differentiation. Id1 may therefore play an early role in anterior-posterior (AP) patterning and mesoderm priming, independent from any effect on overt mesoderm differentiation. This helps to reconcile why BMP is required both for mesoderm differentiation and for the maintenance of pluripotency.

Taken together, our data help to unify the distinct effects of BMP signalling during differentiation of pluripotent cells. BMP maintains high levels of Cdh1, which help to protect the pluripotent state, whilst imposing a posterior identity that ultimately favours mesodermal over neural differentiation.

## Results

### The BMP target gene *Id1* is expressed in the post-implantation pluripotent epiblast

The BMP target gene *Id1* has been reported to inhibit neural induction and instead favour either pluripotency or mesoderm differentiation from pluripotent cells (*Ying et al., 2003a*; *Zhang et al., 2010*),

but the precise events controlled by Id1, and the mechanism by which it acts, are not known. In order to address these questions, we first asked where *Id1* is expressed in the early post-implantation embryo.

It has been reported (*Jen et al., 1997*) that *Id1* is expressed around the embryonic-anembryonic boundary and around the primitive streak of gastrulating mouse embryos, but it is not clear whether *Id1* is expressed within pluripotent epiblast cells prior to their commitment to neural or mesodermal fate, or whether it is restricted to migrating mesodermal cells and to extraembryonic tissues during gastrulation. We decided to use a reporter strategy to examine the expression pattern of *Id1* and its relationship to markers of pluripotency and differentiation during early development.

We generated Id1-Venus (Id1V) reporter cells using a targeting construct designed to express a fusion between Id1 and Venus from the endogenous *Id1* locus (*Nam and Benezra, 2009*) (*Figure 1A*, *Figure 1—figure supplement 1A–E*). This strategy has previously been demonstrated to faithfully report Id1 expression (*Nam and Benezra, 2009*). We used Id1V reporter cells to generate high-contribution chimeras in order to analyse and quantify Id1 expression specifically within the epiblast and its derivatives (ES cells do not contribute to extraembryonic ectoderm in chimeras).

We first examined chimeras made with Id1V ES cells that were constitutively labelled with a fluorescent red mKate2-NLS protein, in order to examine the distribution of Id1V in the early post-implantation embryo. The expression pattern of Venus faithfully recapitulates the expression of *Id1* mRNA in these chimeric embryos (*Figure 1—figure supplement 1F,G*). Id1V is readily detectable at E6.5 throughout the proximal epiblast with expression declining more distally (*Figure 1B*). The highest levels of Id1V therefore mark regions of the epiblast fated for mesoderm and surface ectoderm but not neural fates (*Cajal et al., 2012*; *Li et al., 2013*).

We next generated chimeras using a line of Id1V ES cells that lack the red fluorescent lineage label in order to allow us to use antibody staining to quantify expression of both Nanog and T within individual Id1V+ cells at E6.5 (*Figure 1C–G*). T was used to identify the posterior-most region of the epiblast and the primitive streak (*Rivera-Pérez and Magnuson, 2005*) and Nanog was used to identify pluripotent cells within the posterior half of the epiblast (*Osorno et al., 2012*). The majority (>65%) of Id1V+ cells in the proximal epiblast co-express Nanog, with this proportion increasing to >80% within the posterior regions of the proximal epiblast (*Figure 1D,E*). In contrast, the majority (>80%) of Id1V+ cells do not express T, indicating that they are neither specified nor committed to a mesodermal fate at this stage of development (*Figure 1F,G*). These data indicate that Id1V is expressed within pluripotent epiblast and is not restricted to the primitive streak or to differentiating mesoderm.

We also confirmed that most (>80%) of the T+ cells at this stage co-express Nanog (*Figure 1H,I*), confirming that T marks not only differentiating mesoderm but also pluripotent epiblast cells with a proximal posterior positional identity (*Osorno et al., 2012*).

We conclude that the direct BMP target gene *Id1* is expressed in a subpopulation of Nanog+ epiblast cells prior to overt mesoderm differentiation, in addition to its previously described expression in migrating mesoderm (*Jen et al., 1997*).

## BMP signalling maintains expression of Cdh1 during differentiation of pluripotent cells

We next asked what events are regulated by BMP/Id1 during differentiation of pluripotent cells (*Figure 2A*). We examined the effect of BMP4 on cells cultured in serum-free conditions in the absence of any other exogenous growth factors. This system allows us to monitor the kinetics as well as the direction of differentiation.

We used Sox1-GFP reporter ES cells (*Ying et al., 2003b*) to confirm that BMP4 blocks the upregulation of the early neural marker Sox1 (*Figure 2B*) and permits expression of the primitive streak/early mesoderm marker T (*Figure 2C,D*), as expected (*Ying et al., 2003a*). However, despite expressing T, BMP-treated cells do not progress to overt mesoderm differentiation. Differentiation of epiblast to mesoderm is characterised by the loss of Oct4 and Cdh1, whereas BMP4-treated cells remain uniformly Cdh1+ at least for the first 5 days of BMP treatment (*Figure 2B–D*). Furthermore, in the presence of BMP4, downregulation of *Oct4* is delayed, and markers that characterise EMT and overt mesoderm differentiation are not upregulated within the 5-day time frame of this experiment (*Figure 2D*). As a positive control, we checked that these markers were upregulated within 4 days in control cells that were put through a directed mesodermal differentiation protocol based on monolayer culture on collagen IV in the presence of serum (*Nishikawa et al., 1998*) (*Figure 2D*).

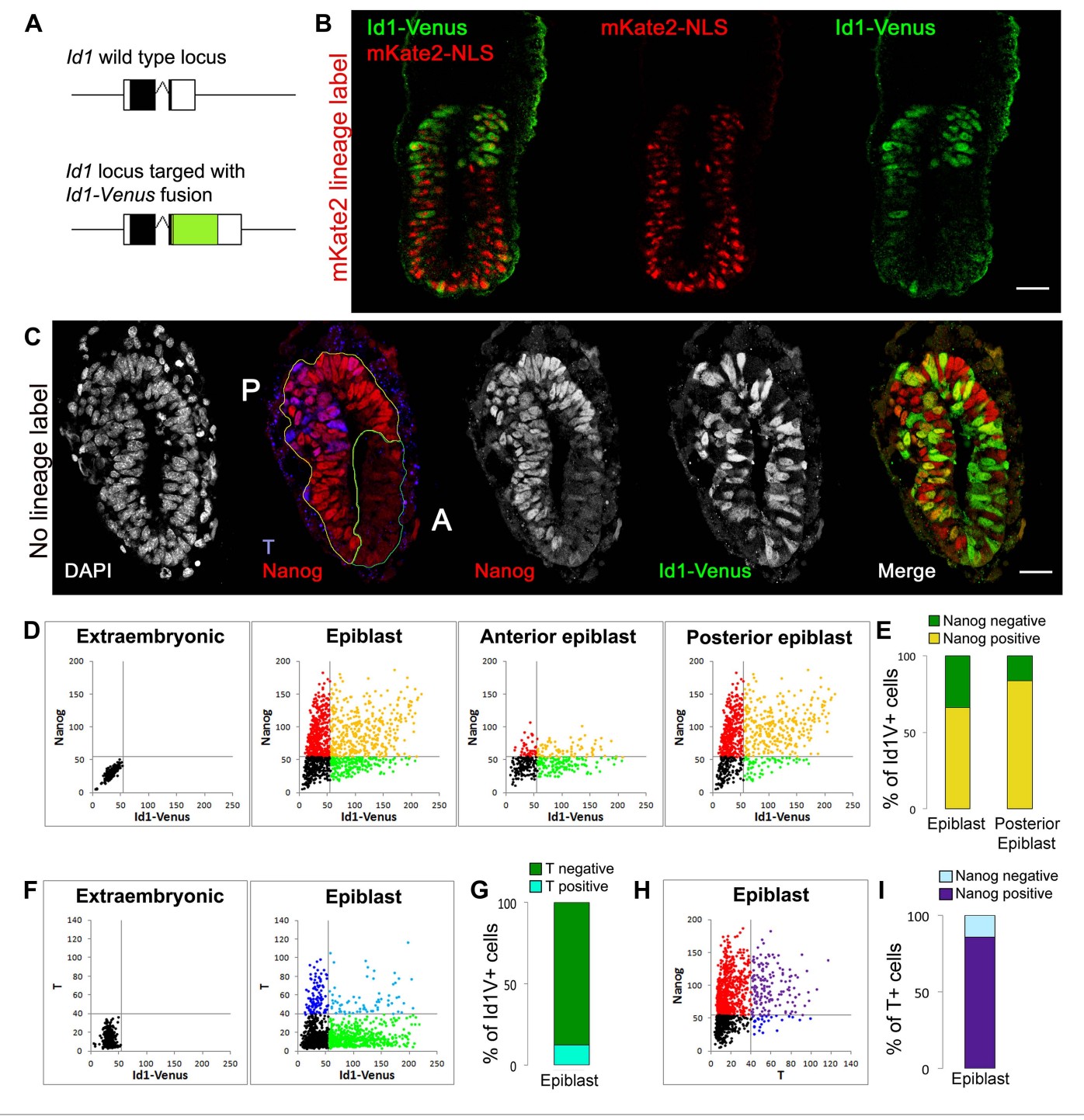

**Figure 1**. The BMP target gene *Id1* is expressed in the pluripotent epiblast. (**A**) Schematic illustrating the genotype of ES cells used for morula aggregation experiments. The coding sequence for an Id1-Venus fusion protein was targeted into one allele of the *Id1* locus. (**B**) Proximal expression of Id1-Venus in E6.5 embryos generated using Id1-Venus ES cells marked with a constitutively expressed red mKate2-NLS lineage label. Scale bar = 30 µm. (**C**) Transverse section of mid-streak stage embryo (E6.5) stained for DAPI, Id1-Venus, Nanog, T. Anterior ('A') and posterior ('P') regions of interest are defined based on expression of Nanog. Scale bar = 30 µm. (**D–I**) Quantification of antibody staining on sections of embryo shown in **C**. Staining of extraembryonic tissue was used as a negative control to define the gates in the dot plots. (**D**) Dot plots representing expression of Id1-Venus and Nanog in single cells in E6.5 epiblast and in the anterior and posterior regions of interest. (**E**) % Nanog-positive cells within the Id1-Venus-positive compartment in E6.5 epiblast. (**F**) Dot plots representing expression of Id1-Venus and T in single cells in E6.5 epiblast. (**G**) % T-positive cells within the Id1-Venus-positive compartment

*Figure 1. Continued on next page*

*Figure 1. Continued*

in E6.5 epiblast. (**H**) Dot plots representing expression of T and Nanog in single cells in E6.5 epiblast. (**I**) % Nanog-positive cells within the T-positive compartment in E6.5 epiblast. See *Figure 1—figure supplement 1* for design and validation of Id1-Venus cells.

The following figure supplements are available for figure 1:

**Figure supplement 1**. Design and validation of Id1V reporter ES cells.

T is first expressed in the posterior proximal epiblast prior to overt mesoderm differentiation (*Rivera-Pérez and Magnuson, 2005*) (see also *Figure 1H,I*) and marks a mesoderm-primed pluripotent state (*Kubo et al., 2004*; *Bernemann et al., 2011*). We therefore considered whether the ability of BMP4 to upregulate T may represent an effect on positional identity within the epiblast rather than a direct effect to drive mesoderm commitment. In support of this idea, we found that in the presence of BMP4, T is coexpressed with Cdh1 (*Figure 2C*). Not all cells persist in a T+ Cdh1+ state: a subpopulation of cells (typically between 10 and 25%) are able to differentiate into a Cdh1+ Tcfap2a (AP2α)+ epithelial cell type similar to early surface ectoderm (*Mitchell et al., 1991*) (*Figure 2C*). This is consistent with the recent finding that exogenous BMP4 promotes surface ectoderm differentiation when added to explants from the E7.0 mouse epiblast (*Li et al., 2013*).

We conclude that BMP4, in the absence of other exogenous signals, imposes a T+ mesoderm-primed posterior positional identity but prevents cells from downregulating Cdh1. BMP-treated cells are delayed in their differentiation and either persist in a Cdh1+ posterior-epiblast like state or ultimately differentiate into Cdh1+ surface-ectoderm-like cells (*Figure 2E*).

## Id1 maintains Cdh1 expression and imposes a mesoderm-primed positional identity

We next asked whether the BMP target gene *Id1* mediates the effects of BMP to maintain Cdh1 expression during differentiation of pluripotent cells.

We generated ES cells in which a Flag-tagged *Id1* transgene is expressed in the presence of doxycyline (dox). We first confirmed that dox-mediated induction of *Id1* is able to block neural induction as expected (*Figure 3—figure supplement 1*). We next tested the effect of Id1 on EMT markers. Under serum-free monolayer conditions in the absence of exogenous growth factors, Id1 delays the loss of *Cdh1* and suppresses the upregulation of EMT markers *Cdh2, Zeb1* and *Zeb2* (*Thiery et al., 2009*) (*Figure 3A*).

We then examined the effect of Id1 on the rate and direction of differentiation. Id1 promotes the upregulation of *T*, a marker of posterior epiblast and primitive streak (*Blum et al., 1992*; *Rivera-Pérez and Magnuson, 2005*), but delays the downregulation of the epiblast markers *Oct4, Fgf5* and *Otx2* (*Figure 3B*). These data are consistent with the idea that Id1, like BMP, favours a posterior proximal epiblast identity whilst delaying overt differentiation. In keeping with this idea, Id1 favours the re-expression of *Nanog*, which marks posterior pluripotent epiblast, and Id1 also reduces expression of *Sox2*, which marks anterior pluripotent epiblast (*Osorno et al., 2012*) (*Figure 3B*). We confirmed that *Nanog* in this context is likely to be marking a posterior epiblast-like cell rather than an earlier naïve pluripotent state because other markers of naïve pluripotency, *Klf4, Rex1* and *Esrrb*, are rapidly lost in Id1 overexpressing cells, as they are in control cells (*Figure 3B,C*).

In further support of the idea that Id1 delays differentiation, we used quantitative immunofluorescence to demonstrate that in the presence of exogenous Id1, T+ cells remain predominantly Cdh1+ and >90% of them coexpress Oct4 (*Figure 3D,E*). These T+ cells are therefore unlikely to be differentiated mesoderm cells and are more likely to be equivalent to the T+ cells in the proximal epiblast of the early embryo that have not yet entered the primitive streak (*Rivera-Pérez and Magnuson, 2005*).

Dox treatment does not entirely block differentiation (*Figure 3—figure supplement 1C*). Neurons begin to emerge in the dox-treated cultures after 5 days. However, these neurons do not express the Flag-tagged Id1 transgenic protein. Indeed, around 50% of cells are negative for Flag on day 5 (*Figure 3—figure supplement 1D*). Therefore, in order to address whether Id1 is sufficient to block differentiation, we turned to a constitutive expression system. We generated clonal lines of ES cells stably overexpressing *Id1* under the control of the ubiquitously expressed (*Niwa et al., 1991*) CAGS promoter. In these cell lines, *loxP* sites flank the coding sequence for *Id1* and upon Cre-mediated excision a downstream *GFP* cassette is expressed. We first confirmed that Id1 is able to block neural

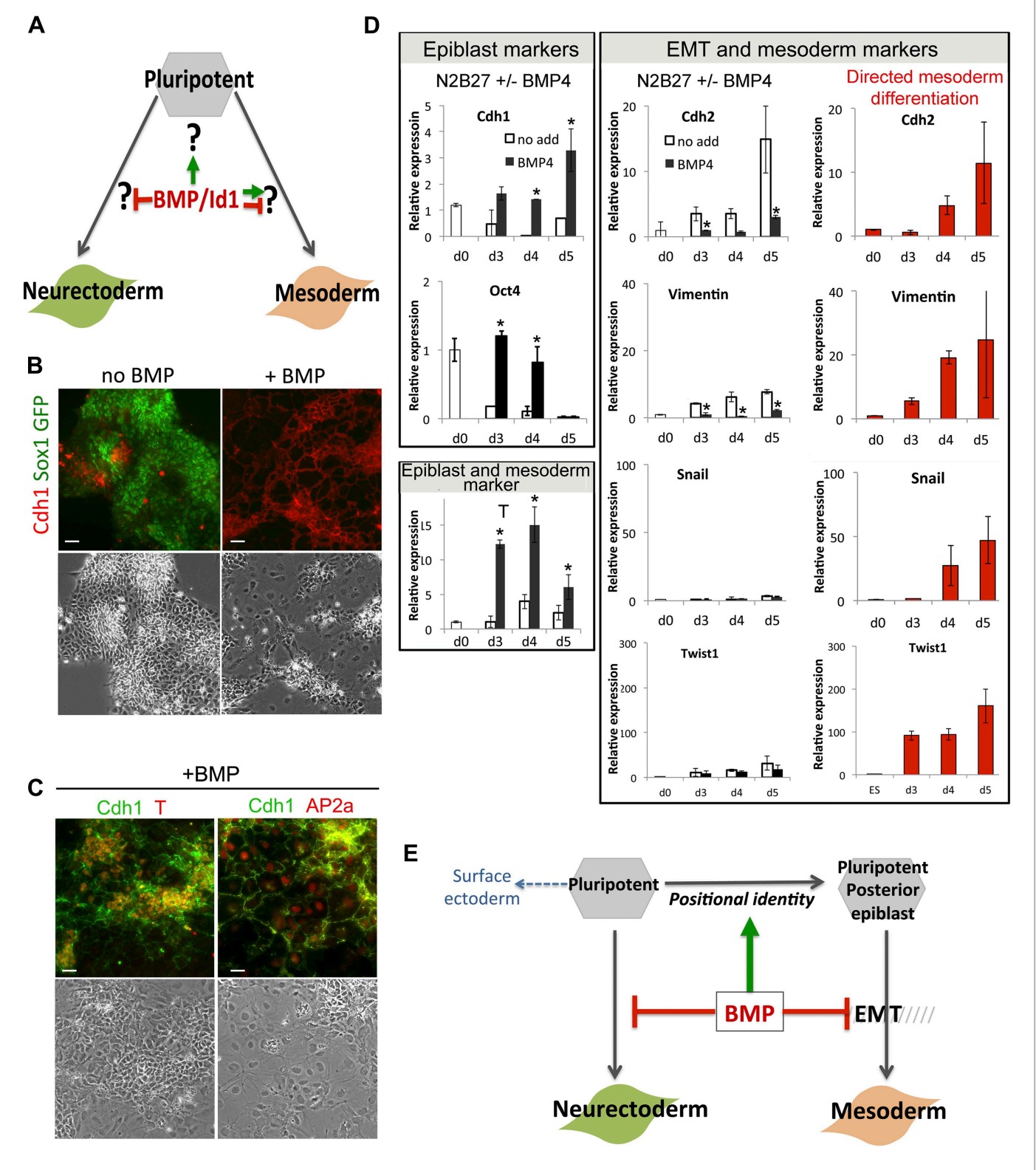

**Figure 2**. BMP signalling maintains Cdh1 expression during differentiation of pluripotent cells. (**A**) BMP blocks neural differentiation, maintains pluripotency and has both positive and negative effects on mesoderm differentiation, but the mechanisms by which BMP acts in each case are not well understood. (**B**) Sox1-GFP ES cells were placed under neural differentiation conditions for 4 days in the presence or absence of 5 ng/ml BMP and stained for markers

*Figure 2. Continued on next page*

*Figure 2. Continued*

as shown. Scale bar = 30 μm. (**C**) E14tg2α ES cells were placed under neural differentiation conditions for 4 days in the presence of 5 ng/ml BMP and stained for markers as shown. Scale bar = 30 μm. (**D**) qRT-PCR analysis of ES cells placed under neural differentiation conditions in the presence or absence of 5 ng/ml BMP, or under directed mesoderm differentiation conditions. (**E**) We propose that BMP imposes a posterior positional identity whilst delaying both EMT and overt mesoderm differentiation but permitting surface ectoderm differentiation. qRT-PCR data are represented as mean +/− standard deviation and * indicates p<0.05 relative to no-BMP4 control.

induction as expected, and that this phenotype can be reverted by Cre-mediated excision of the *Id1* transgene (*Figure 3—figure supplement 1E–H*). Unlike the dox-inducible system, this constitutive expression system is sufficient to completely block neurogenesis throughout the culture (*Figure 3—figure supplement 1G*).

Constitutive Id1 expression is sufficient to delay the downregulation of *Oct4* (*Figure 3—figure supplement 2*) and to block neurogenesis (*Figure 3—figure supplement 1G*). However, it is not sufficient to completely block differentiation. Id1-overexpressing cells eventually differentiate predominantly into mesoderm rather than neurectoderm or endoderm, even under neural differentiation conditions (*Figure 3—figure supplement 2*). In this regard, exogenous Id1 does not recapitulate the effect of exogenous BMP4 described in *Figure 2*. The ability of Id1-overexpressing cells to generate mesoderm at later time points under these conditions may reflect the progressive downregulation of the *Id1* transgene (*Figure 3—figure supplement 1*), or it may reflect an Id1-independent effect of BMP4 on the later stages of differentiation.

Taken together, these data suggest that the BMP target gene *Id1* does not drive overt mesoderm differentiation but rather maintains Cdh1 and helps to maintain a primed pluripotent state similar to that of the post-implantation epiblast, whilst imposing a proximal posterior (T+ Nanog+) epiblast identity that is primed for mesoderm differentiation.

## Cdh1 is downregulated in synchrony with Oct4 during neural differentiation

The data described above reveal an unexpected ability of BMP to maintain Cdh1 expression. We wondered how this may relate to the ability of BMP to block neural induction, given that Cdh1 is downregulated during the early stages of neural tube formation *in vivo* (*Hatta and Takeichi, 1986*; *Aaku-Saraste et al., 1996*; *Karpowicz et al., 2007*) and during neural differentiation of ES cells (*Sterneckert et al., 2010*; *Kamiya et al., 2011*).

We first asked whether the loss of Cdh1 occurs subsequently to neural differentiation or whether it is tightly associated with the differentiation process itself. Since the rate of neural differentiation varies considerably between individual cells, it was important to perform this analysis at the single cell level. We therefore performed single cell quantitative RT-PCR in order to monitor the transcriptional downregulation of *Cdh1*, as cells proceed from pluripotency towards neural differentiation. We confirmed that *Cdh1* is largely absent from cells that have upregulated early neural markers (*Figure 4A*). Strikingly, we also observed a close correlation between the expression of *Cdh1* and *Oct4* at all stages of the differentiation process (*Figure 4B*), consistent with the idea that *Cdh1* is downregulated at precisely the time that cells transit from Oct4+ epiblast towards neurectodermal differentiation. We confirmed a similar correlation between loss of Oct4 and downregulation of Cdh1 in vivo: Cdh1 and Oct4 protein expression decline in synchrony within the developing neuroepithelium (*Figure 4C,D*).

We confirmed these results using Sox1-GFP neural reporter ES cells (*Ying et al., 2003b*) to monitor early neural differentiation under uniform monolayer differentiation conditions. Under these conditions, around 20–40% of cells fail to become Sox1-GFP+ after 4 days; around half of these persist as undifferentiated ES cells whilst the other half differentiate into non-neural (Sox1-negative) cell types (*Ying and Smith, 2003*; *Lowell et al., 2006*). The reasons for this heterogeneity in differentiation capacity are not fully understood. We observed that the subpopulation of cells that fail to respond appropriately to a neural environment by upregulating Sox1-GFP also fail to downregulate Cdh1, either remaining as Oct4+ Cdh1+ undifferentiated cells or differentiating into Cdh1+ Krt8+ epithelial cells (*Figure 4E,F*).

We conclude that Cdh1 is downregulated in synchrony with Oct4 during neural differentiation, and that heterogeneity in neural differentiation capacity within populations of ES cells is associated with a failure to downregulate Cdh1 in a subpopulation of those cells.

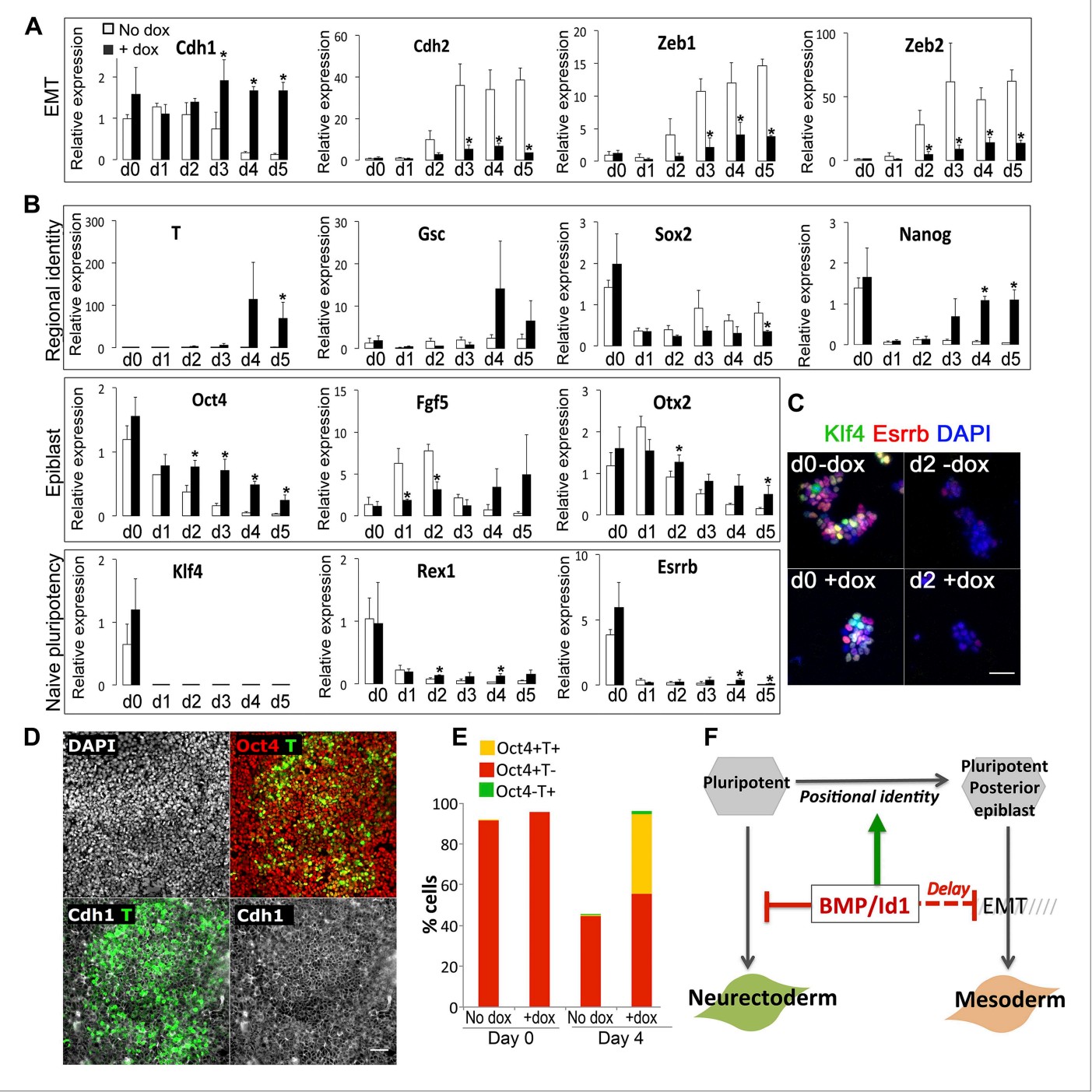

**Figure 3**. Id1maintains Cdh1 expression and imposes a mesoderm-primed positional identity. (**A** and **B**) qRT-PCR analysis of dox-inducible Id1 ES cells placed under neural differentiation conditions (N2B27 alone) in the presence or absence of dox. (**C**) Dox-inducible Id1 ES cells placed under neural differentiation conditions (N2B27 alone) in the presence or absence of dox and stained for markers as shown. Scale bar = 50 μm. (**D**) Dox-inducible Id1 ES cells on day 4 under neural differentiation conditions (N2B27 alone): T is coexpressed with Cdh1 and Oct4. Scale bar = 50 μm. (**E**) Quantification of antibody staining of dox-inducible Id1 ES cells in the presence or absence of dox on day 0 and day 4 under neural differentiation conditions (N2B27 only). A minimum of 750 nuclei were scored for d0 samples and a minimum of 7000 nuclei were scored for d4 samples. (**F**) We propose that the BMP target gene *Id1* imposes a posterior positional identity whilst delaying both EMT and overt mesoderm differentiation. qRT-PCR data are represented as mean +/− standard deviation and * indicates p<0.05 relative to no-dox control. See *Figure 3—figure supplements 1 and 2*.

The following figure supplements are available for figure 3:

*Figure 3. Continued on next page*

*Figure 3. Continued*

**Figure supplement 1**. Characterisation of Id1 overexpressing lines.

**Figure supplement 2**. Id1 overexpressing cells differentiate preferentially into mesoderm rather than neuroectoderm or endoderm.

## Downregulation of Cdh1 is a limiting requirement for neural differentiation

The data above confirm that the earliest stages of neural differentiation are accompanied by the downregulation of Cdh1. We next asked whether this is simply a passive accompaniment to neural differentiation, or whether it is a limiting requirement, playing an active role in allowing cells to progress towards neural differentiation.

Having confirmed that a subpopulation of cells fail to downregulate Cdh1 under neural differentiation conditions, and that these cells also fail to differentiate into neural cells (*Figure 4E,F*), we next asked whether it is possible to rescue neural differentiation in this recalcitrant subpopulation by experimentally suppressing Cdh1 activity.

We used function blocking antibodies (*Yoshida-Noro et al., 1984*) in order to abolish the adhesive abilities of Cdh1. Blocking antibodies were added to Sox1-GFP reporter cells for the first 48 hr of neural differentiation and then removed, whereupon the cells rapidly regain normal cell-cell contacts. 1 day later, a minority of control cells are just starting to express Sox1-GFP, whereas the antibody-treated cells show strong Sox1-GFP expression throughout the culture. By the fourth day of differentiation, the antibody-treated cultures are almost uniformly Sox1-GFP+ (*Figure 5A,B*). Flattened non-neural differentiated cells are readily apparent in control cultures, but these are absent from antibody-treated cultures (*Figure 5A*). qRT-PCR analysis confirmed that early transient exposure to the blocking antibody accelerates the loss of the pluripotency marker *Oct4* and the gain of the neural marker *Sox1* (*Figure 5C*). This indicates that transient loss of Cdh1 activity at the very start of the differentiation process is sufficient to drive ES cells efficiently towards neural differentiation.

As an alternative loss-of-function approach, we made use of C*dh1* null ES cells (*Larue et al., 1996*). These cells do not form adherens junctions but do self-renew and retain multilineage differentiation potency. *Cdh1* null ES cells differentiate into neural cells more rapidly and efficiently in comparison with parental controls (*Figure 5D*).

We sought to confirm these data using an alternative robust and regulatable system for suppressing Cdh1 activity. The transcription factor Twist1 is able to downregulate Cdh1 when bound to its heterodimerisation partner E47 (*Thiery et al., 2009*). We engineered a construct designed to express Twist1 tethered by a flexible linker to E47: we refer to this heterodimeric fusion protein as TwistE. We confirmed that TwistE efficiently downregulates *Cdh1* when expressed in ES cells by episomal transfection (*Figure 5—figure supplement 1A,B*). We next generated ES cells lines that allowed dox-inducible expression of TwistE (*Figure 5—figure supplement 1C,D*) and made use of these dox inducible TwistE cell lines to suppress Cdh1 expression during neural differentiation.

We reasoned that if downregulation of Cdh1 is a limiting requirement for neural induction, activation of TwistE should be able to enhance the efficiency of this process. This is indeed the case: activation of TwistE has a striking ability to accelerate neural differentiation. Cells that are treated with dox generate extensive networks of neurons after only 3 days of exposure to neural conditions, whilst untreated control sister dishes do not start to generate significant numbers of neurons until day 5 (*Figure 5E,F*). Neural markers *Sox1, Zfp521, Zeb2, Blbp, Sox9, Mash1 and Math1* are upregulated prematurely and *Oct4* is lost more quickly in dox-treated cultures (*Figure 5G*). Activating TwistE only for the first day of differentiation is sufficient to accelerate the early neural programme (*Figure 5H*), indicating that Cdh1 activity is normally a limiting factor at an early stage in the differentiation process.

Taken together, these observations suggest that downregulation of Cdh1 is a limiting requirement for pluripotent cells to differentiate into neural cells.

## Suppression of Cdh1 rescues the BMP-imposed block to differentiation

The data reported above indicate that BMP/Id can maintain Cdh1, and that downregulation of Cdh1 is a limiting event during differentiation of ES cells. These findings raise the possibility that the mechanism by which BMP blocks differentiation could be through maintenance of Cdh1. In order to test

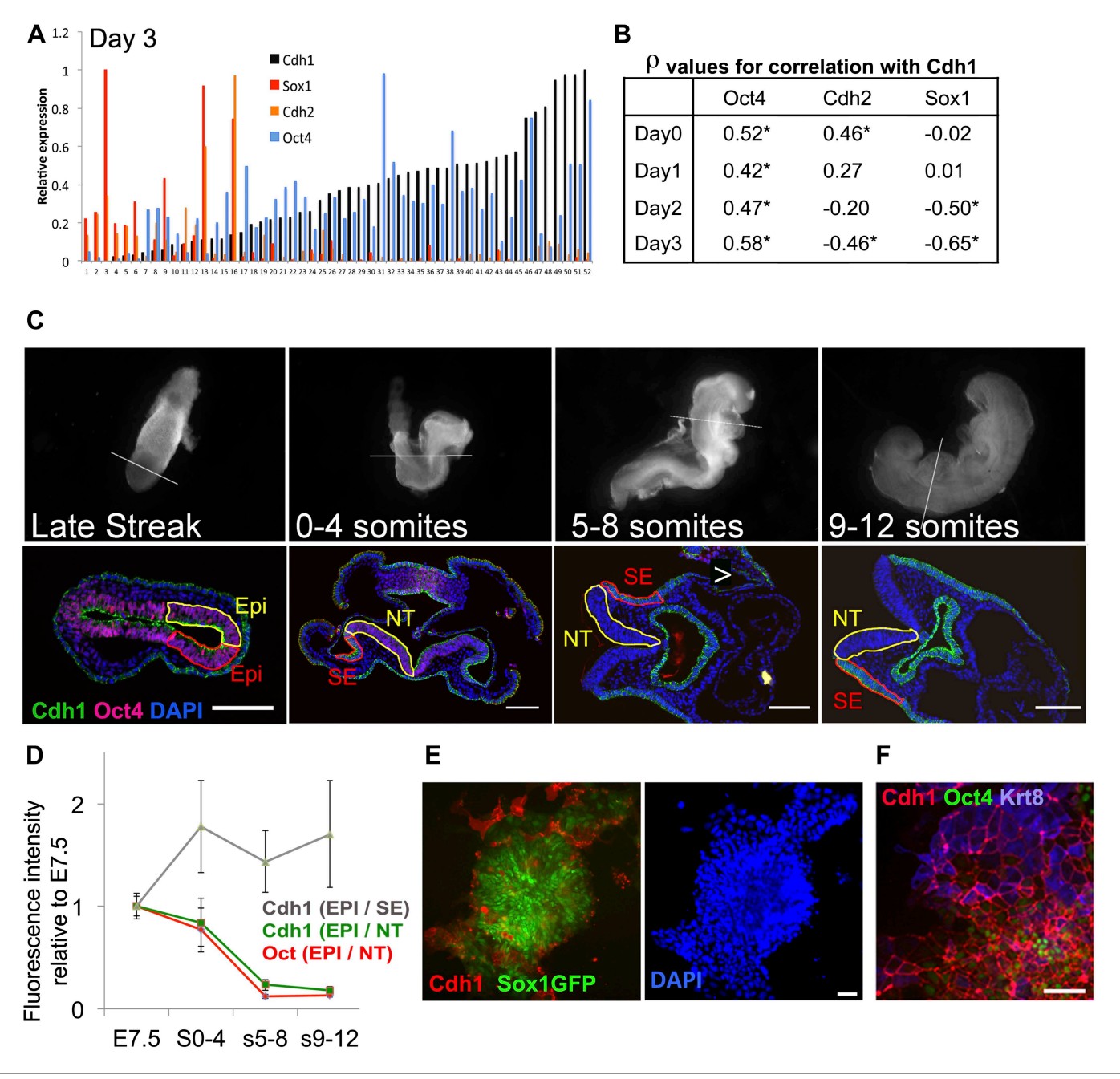

**Figure 4**. Cdh1 is downregulated in synchrony with Oct4 during neural differentiation. (**A** and **B**) ES cells were placed under neural differentiation conditions. 40–80 single cells were analysed by qRT-PCR from the starting population (Day 0) or at daily time points during differentiation. A shows expression levels of *Cdh1*, *Cdh2*, *Oct4* and *Sox1* in single cells analysed at day 3 of differentiation. The expression values for each factor are normalised to the maximum expression level found in this set of cells. The samples are sorted in ascending order of *Cdh1* expression. B shows the Spearman's rank correlation coefficients (ρ) between the named genes and *Cdh1* at each time point. * indicates asymptotic p value <0.05. (**C**) Oct4 and Cdh1 staining in 7 µm sections of embryos at E7.5 (late streak stage) or staged according to number of somite pairs (s) as indicated. Top row shows bright field images of embryos before sectioning with a white line indicating the location of the section shown in the bottom row. Scale bars = 100 µm. NT: Neural tube (Yellow region in later stage embryos), SE: Surface ectoderm (red region in later embryos). (**D**) Quantification of the average intensities of Oct4 and Cdh1 in different structures of the embryo at each stage. EPI: Epiblast (yellow region and red region in LS embryo), Error bars represent the standard deviation of intensities in three different sections. Abbreviations are as described for **C**. (**E**) Sox1-GFP ES cells after 4 days in neural differentiation conditions stained for Cdh1 (red) and DAPI (blue) with Sox1-GFP in green. Scale bar = 30 µm. (**F**) E14tg2α ES cells after 4 days under neural differentiation conditions stained for Cdh1 (red) Keratin8 (blue) and Oct4 (green).Scale bar = 30 µm.

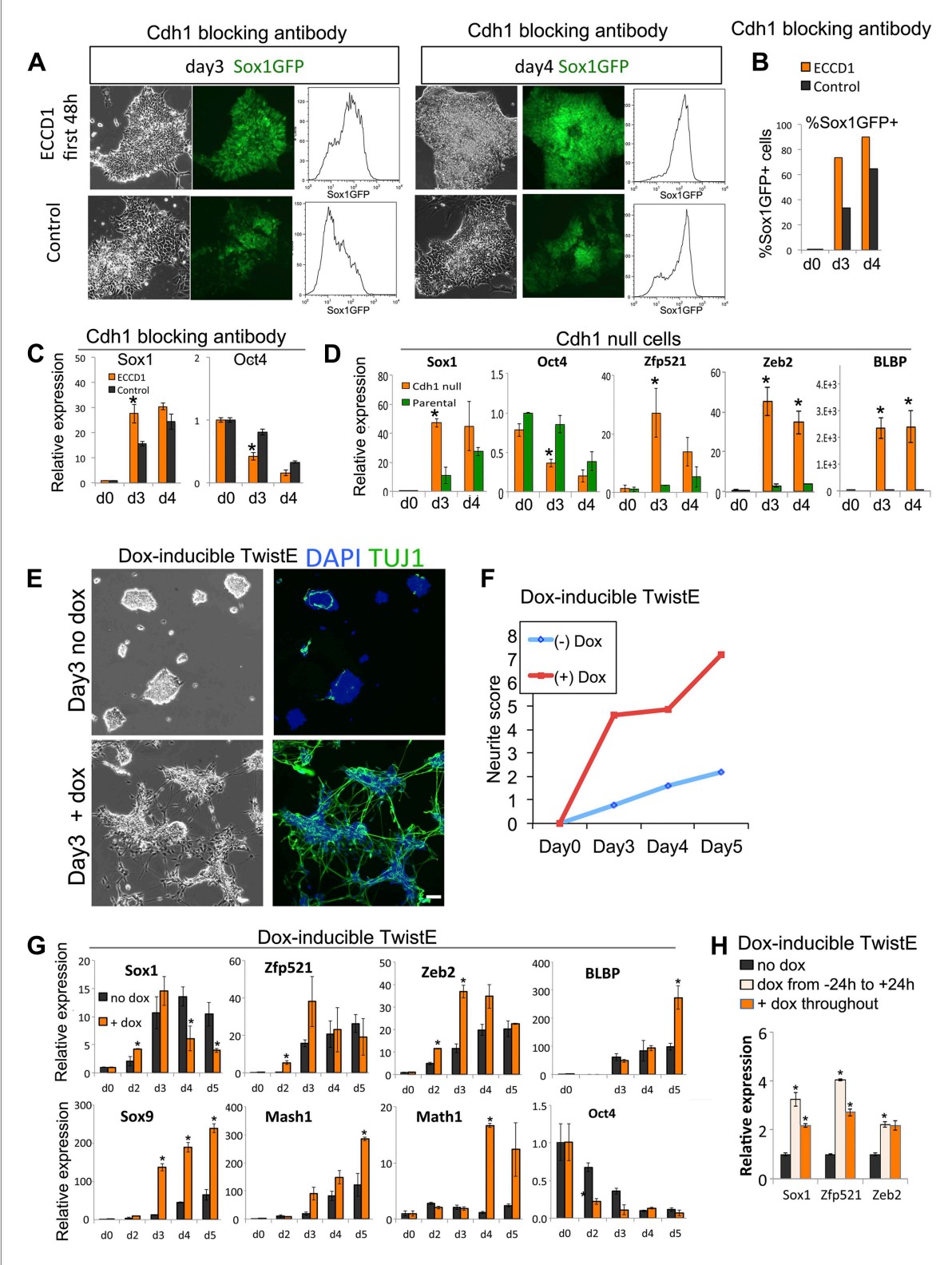

**Figure 5**. Loss of Cdh1 activity is a limiting requirement for neural differentiation. (**A–C**) FACS (**A** and **B**) and qRT-PCR (**C**) analysis of Sox1-GFP ES cells undergoing neural differentiation after exposure to Cdh1 blocking antibody ECCD1 for the first 48 hr of differentiation. Control indicates sister dishes not exposed to ECCD1. (**D**) qRT-PCR analysis of *Cdh1* null or parental D3 ES cells undergoing neural differentiation. (**E** and **F**) Dox-inducible TwistE cells during neural differentiation in the presence or absence of dox. Scale bar = 30 μm. (**F**) shows neurite frequency scored from the experiment illustrated in **E**.
*Figure 5. Continued on next page*

*Figure 5. Continued*

Neurites were scored as described in 'Materials and methods'. (**G**) qRT-PCR analysis of dox-inducible TwistE cells placed under neural differentiation conditions in the presence or absence of dox. (**H**) qRT-PCR analysis of dox-inducible TwistE cells on day 2 of neural differentiation in the absence of dox, the presence of dox throughout the experiment, or after removing dox 24 hr after initiating neural differentiation. qRT-PCR data are represented as mean +/− standard deviation and * indicates p<0.05 relative to no-antibody control, parental control, or no-dox controls. See also *Figure 5—figure supplement 1*.

The following figure supplements are available for figure 5:

**Figure supplement 1**. Design and validation of TwE cell lines.

this directly, we asked whether suppression of Cdh1 is sufficient to rescue the differentiation block imposed by BMP4.

1 ng/ml of BMP4 is sufficient to almost entirely inhibit neural induction and to maintain *Cdh1* expression (*Figure 6A*), as expected. Strikingly, suppression of Cdh1 via dox-induction of TwistE is sufficient to rescue neural differentiation in the presence of 1 ng/ml BMP. Tuj1+ neurons can be readily found in dox-treated cultures but are only rarely found in control sister dishes (*Figure 6A*), and dox treatment rescues the expression of early neural markers *Mash1, Zfp521, Sox9* and *Pax3* to levels similar to those observed in the absence of exogenous BMP (*Figure 6B*). *Sox1* is downregulated before day 4, in keeping with the ability of TwistE to accelerate the early stages of the differentiation process (*Figure 5G,H* and data not shown). TwistE also abolishes the ability of BMP4 to sustain the expression of epiblast markers *Oct4, Cdh1, Fgf5* and *T* (*Figure 6B*). We conclude that under these conditions BMP acts primarily through Cdh1 to maintain the expression of epiblast markers and block the expression of neural markers.

We noticed that the dox-treated cultures, in addition to containing abundant neurons (*Figure 6A*), also contain many Tuj1-negative cells that do not have a neural morphology (data not shown). This suggests that TwistE does not exclusively drive neural differentiation in this context but may be rescuing a more general delay to differentiation into multiple lineages. We speculated that BMP might have two independent effects: one to block multilineage differentiation by maintaining Cdh1 and another to influence the direction of differentiation independently of Cdh1.

In order to test this idea, we increased the dose of BMP to 5 ng/ml, with the aim of imposing a posterior positional identity that favours mesodermal and endodermal rather than neural fates (*Figure 2*). Under these conditions, activation of TwistE eliminates Cdh1 expression from the majority of cells and upregulates the mesenchymal and neural marker Vimentin (*Figure 6C*), but we did not observe any Tuj1+ neurons either in the presence or the absence of dox treatment (data not shown). We next confirmed that 5 ng/ml BMP4 upregulates posterior epiblast markers whilst delaying overt differentiation into mesoderm and endoderm (*Figure 6D*, black bars), as expected. Under these conditions, suppression of Cdh1 via dox-induction of TwE efficiently rescues mesoderm and endoderm differentiation, but not neural differentiation (*Figure 6D*).

We conclude that part of the mechanism by which BMP helps to maintain pluripotency is through maintenance of Cdh1, which interferes with efficient neural and mesodermal differentiation. Higher levels of BMP also impose a posterior identity that primes cells for mesoderm and is incompatible with a neural fate. Taken together, these findings help to explain why BMP can at the same time protect pluripotency and promote mesoderm differentiation.

## Discussion

Changes in the expression of the cell-cell adhesion molecule E-Cadherin (Cdh1) are often associated with changes in cell identity throughout development and into adult life (*Thiery et al., 2009*). For example, downregulation of Cdh1 is associated with the epithelial-mesenchymal transition (EMT) that accompanies the formation of mesoderm at the primitive streak and the emergence of neural crest cells from the dorsal neural tube (*Thiery et al., 2009*). The formation of the early neural plate from the epiblast does not bear all the hallmarks of a classical EMT, but it is accompanied by loss of Cdh1 and gain of N-Cadherin (Cdh2) and Vimentin (*Hatta and Takeichi, 1986*; *Aaku-Saraste et al., 1996*). There is already evidence that EMT is required for overt mesoderm differentiation (*Martínez-Estrada et al., 2010*), and we find here that downregulation of Cdh1 is similarly a limiting event for neural differentiation.

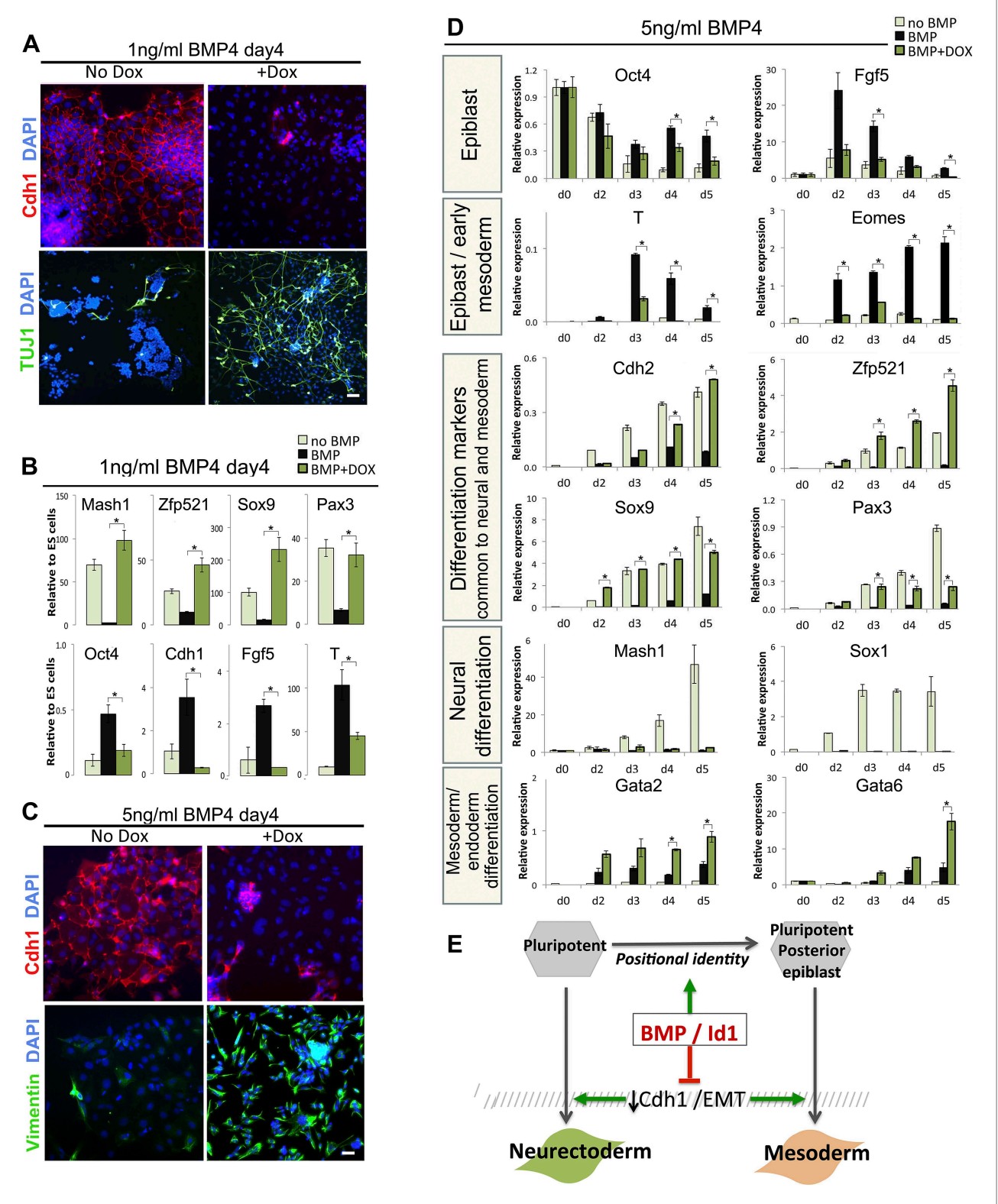

**Figure 6**. Suppression of Cdh1 rescues the BMP-imposed delay to differentiation. (**A**) Antibody staining for the indicated markers in dox-inducible TwistE cells after 4 days under neural differentiation conditions in the presence of 1 ng/ml BMP4. Scale bar = 30 μm. (**B**) qRT-PCR analysis of dox-inducible TwistE cells after 4 days under neural differentiation conditions in the presence or absence of 1 ng/ml BMP4. (**C**) Antibody staining for the indicated markers in dox-inducible TwistE cells after 4 days under neural differentiation conditions in the presence of 5 ng/ml BMP4. (**D**) qRT-PCR analysis of

*Figure 6. Continued on next page*

*Figure 6. Continued*

dox-inducible TwistE cells under neural differentiation conditions in the presence or absence of 5 ng/ml BMP4. (**E**) MODEL: we propose that BMP controls differentiation of pluripotent cells by suppressing the downregulation of Cdh1. qRT-PCR data are represented as mean +/− standard deviation and * indicates p<0.05 relative to control samples treated with BMP but not treated with dox.

Although it is well established that BMP signalling contributes to the maintenance of pluripotency (*Ying et al., 2003a*; *Di-Gregorio et al., 2007*), the underlying mechanisms remain unclear. We propose that BMP suppresses differentiation by maintaining expression of Cdh1. We support this idea by showing that suppressing Cdh1 is sufficient to rescue the BMP-imposed block to multilineage differentiation.

It is also unclear how BMP signalling favours mesodermal over neural fates. We find that the BMP target gene *Id1* imposes a proximal posterior identity on undifferentiated cells, priming them towards mesoderm. The ability of Id1 to block neural differentiation is therefore unlikely to be explained entirely by inhibition of a neural-inducing transcription factor as is commonly assumed, nor by a direct activation of mesoderm differentiation, but rather to a shift in the positional identity of pluripotent cells that occurs in the absence of overt differentiation. This may explain why, to our knowledge, no Id-regulated pro-neural bHLH transcription factor has been found that explains the ability of Id1 to influence neural induction.

*Id1* null ES cells have a defect in the maintenance of pluripotency (*Romero-Lanman et al., 2011*), consistent with our finding that Id1 delays exit from an epiblast-like state. We note, though, that the *Id1* null phenotype is also likely to reflect an earlier pre-implantation role for Id1: to suppress the activity of Tcf15 (*Davies et al., 2013*). The consequence of this would be to maintain expression of Nanog and suppress the transition to a T+ epiblast-like state (*Davies et al., 2013*), in keeping with the *Id1* null ES cell phenotype (*Romero-Lanman et al., 2011*).

The two effects of BMP, to maintain Cdh1 and to impose a posterior identity, can be uncoupled by varying the dose of BMP. Lower doses of BMP are sufficient to maintain Cdh1 and to suppress neural fate, but insufficient to impose a stable posterior identity. Strikingly, suppressing Cdh1 at these lower doses of BMP is sufficient to rescue neural differentiation, indicating that the primary mechanism by which BMP inhibits neural differentiation in this context is through its ability to control Cadherin activity. This is an unexpected finding that highlights the importance of cell adhesion and tissue organisation in controlling receptivity of cells to differentiation cues (*Stepniak et al., 2009*; *Faunes et al., 2013*).

There is a growing body of evidence that Cdh1 influences growth factor receptiveness of ES cells, and that it can consequently stabilise particular pluripotent states (*Chou et al., 2008*; *Soncin et al., 2009*; *Li et al., 2010*; *Redmer et al., 2011*; *Hawkins et al., 2012*; *del Valle et al., 2013*; *Faunes et al., 2013*), but it is not known what controls Cdh1 expression in this context. We fill this gap with our finding that BMP/Id maintains Cdh1 expression during differentiation of ES cells. Interestingly, *Id1* is excluded from the anterior streak in vivo (*Jen et al., 1997*), so it is possible that it plays a more general role in regulating EMT in the early embryo.

In vivo, the BMP signals that are generated from the extraembryonic ectoderm not only block neural induction but are also required for formation of mesoderm (*Mishina et al., 1995*; *Winnier et al., 1995*; *Lawson et al., 1999*; *Beppu et al., 2000*; *Davis et al., 2004*). However, BMP signalling is also necessary for maintenance of pluripotency (*Di-Gregorio et al., 2007*), and mouse embryos in which BMP has been experimentally overactivated have an impairment in mesoderm differentiation that cannot easily be explained by models in which BMP drives mesoderm differentiation (*Aoyama et al., 2012*). Our findings unite these observations by showing how BMP can prime cells for mesoderm fate by imposing a posterior identity, but restrain them from undergoing overt mesoderm formation by suppressing EMT.

In frogs, antagonism of BMP by the organiser induces anterior-posterior axis formation. It has been proposed (*Constance Lane et al., 2004*) that BMP antagonism acts by stimulating mediolateral intercalation of caudal progenitors, releasing them to participate in the formation of the axis, and that the normal role of BMP is to preserve a reservoir of progenitors throughout axis elongation. We propose that BMP acts by a similar mechanism during differentiation of pluripotent mouse cells, preserving a population of undifferentiated cells by preventing the loss of Cdh1. The mechanism by which BMP signalling controls EMT will be an interesting area for future study. This may be connected with BMP's ability to antagonise Nodal signalling, given that Nodal is required for the initiation of EMT at the primitive streak (*Arnold and Robertson, 2009*; *Yan et al., 2009*).

Our work also has implications for controlling uniform differentiation of mouse ES cells. It has been known for some time that a subpopulation of ES cells tends to resist neural differentiation, but the reasons for this are not clear. We propose that this subpopulation of cells fail to respond appropriately to a neural-inducing environment because they are unable to downregulate Cdh1. We demonstrate this directly by showing that experimental suppression of Cdh1 activity can rescue neural differentiation capacity in this recalcitrant subpopulation resulting in a more homogenous differentiation response. Importantly, this can be achieved by non-genetic means, such as function-blocking E-Cadherin antibodies, providing a convenient and readily applicable means to improve our control over pluripotent cells.

In summary, we have identified the loss of Cdh1 activity as a limiting event for the earliest stages of neural differentiation and we propose that BMP controls differentiation in part by maintaining Cdh1 expression. The strategy of coupling changes in cell adhesion to transcriptional control of cell fate specification is used several times during development as a way of coordinating tissue morphogenesis (*Owens et al., 2000*; *Martínez-Estrada et al., 2010*; *Watt and Fujiwara, 2011*), and it will be interesting to investigate the implications of our findings for germ-layer formation in the early embryo.

## Materials and methods

### Mouse ES cell culture

ES cells were maintained in GMEM supplemented with 2-mercaptoethanol, non-essential amino acids, glutamine, pyruvate, 10% foetal calf serum (FCS) and 100 units/ml LIF on gelatinised tissue culture flasks (*Smith, 1991*).

### Cell lines

To generate dox-inducible *Id1*-overexpressing ES cells, E14tg2α ES cells were transfected with a plasmid containing a CAGS promoter driving expression of the reverse tetracycline-controlled transactivator (rtTA), followed by an IRES and a Blasticidin S resistance gene. The cells were cultured in the presence of 20 µg/ml Blasticidin S for two passages. The DNA sequence corresponding to N-terminally 3 × Flag-tagged mouse *Id1-001* was generated by overlapping extension PCR. This sequence was subcloned downstream of a tet-responsive promoter in a plasmid containing a *Pgk* promoter driving expression of a Hygromycin B resistance gene. This construct was transfected into the E14tg2α ES cells under Blasticidin S selection. The cells were subject to dual selection in 20 µg/ml Blasticidin S and in 200 µg/ml Hygromycin B prior to clone isolation and expansion.

To generate *Id1*-overexpressing ES cells, E14tg2α ES cells were transfected with a previously described *Id1*-overexexpression plasmid (*Ying et al., 2003a*), containing a CAGS promoter driving the expression of *Id1* cDNA, followed by an IRES, a Puromycin resistance gene (*Pac*) and a polyadenylation signal sequence (pA). *Id1-IRES-Pac-pA*are flanked by *loxP* sites. Downstream of the second *loxP* site there is a *GFP* transgene, not expressed in *Id1*-overexpressing cells but activated after Cre-mediated excision of *Id1-IRES-Pac-pA*. Revertant cells were obtained by transfecting *Id1*-overexpressing cells with a circular plasmid encoding Cre recombinase and screening for expression of GFP.

To generate Id1-Venus reporter lines, E14tg2α cells were electroporated with a targeting construct provided by H-S Song and R Benezra (Sloan-Kettering Memorial Cancer Centre, NY, USA) designed to express a fusion between Id1 and Venus from the endogenous *Id1* locus. This construct is described in detail in *Nam and Benezra (2009)*. Correct targeting was determined by Southern blotting following the strategy described in *Nam and Benezra (2009)*.

To generate a fluorescent lineage label, the sequence encoding the mKate2 red fluorescent protein was fused at its 3′ end prior to its stop codon with three repeats of the SV40 nuclear localisation signal sequence (NLS). This sequence was subcloned downstream of a CAGS promoter and followed by an IRES and *Pac*. This construct was transfected into the Id1-Venus reporter cells and clones were isolated following Puromycin selection.

Chimeric embryos were generated by morula aggregation.

To generate dox-inducible *Twist1-E47* ES cell lines: the DNA sequence encoding an N-terminally Flag-tagged mouse Twist1 tethered at its C-terminus to the N-terminus of mouse E47 through a 13 amino acid flexible linker of sequence TGSTGSKTGSTGS was generated by overlapping extension PCR. This sequence was placed under a tet-responsive promoter and transfected into the E14tg2A_AW2 cell line, which contains the coding sequence for the reverse tetracycline-controlled

transactivator (*rtTA*) integrated into the *Rosa26*locus and expressed from the *Rosa26* promoter. The E14tg2A_AW2 cell line is described in more detail elsewhere (*Zhou et al., 2013*). Cells were plated at clonal density and clones expanded in ES cell culture conditions under Hygromycin B selection. The expression of Flag-Twist1-E47 was tested in a number of clones through qRT-PCR analysis (using primers within *Twist1*) and western blot analysis (probed by anti-Flag antibody) over a 5-day time course of treatment in the presence or absence of doxycycline (dox). A dox-inducible Twist1-E47 cell line was selected that had negligible leakiness of Twist1-E47 in the absence of dox, and moderate overexpression in the presence of dox. Cell lines for dox-inducible expression of monomeric N-terminally flag tagged Twist1 were generated in the same way.

## Differentiation protocols

Monolayer neural differentiation is described in detail in *Pollard et al. (2006)*. Briefly, ES cells were washed to remove all traces of serum and then plated onto gelatin-coated tissue culture plastic in N2B27 serum-free medium. N2B27 consists of a 1:1 ratio of DMEM/F12 and Neurobasal media supplemented with 0.5% modified N2 (made in house as described in *Pollard et al., 2006*), 0.5% B27 and 2- mercaptoethanol. Medium was changed every second day. Directed mesoderm differentiation on collagen IV was carried out as previously described (*Nishikawa et al., 1998*).

## qRT-PCR

Primers used for qRT-PCR are described in *Supplementary file 1*. All expression values are normalised to those of the housekeeping gene *TBP*.

## Statistical analysis

Experiments in *Figure 6D* were performed in four replicates, with paired replicates performed on different days. All other experiments were performed on three independent occasions. Statistical significance was calculated using a paired Student's *t* test.

## Single cell qRT-PCR

RNA reverse transcription and cDNA pre-amplification from single cells were performed as previously described (*Dalerba et al., 2011*) with some modifications. Each well of a 96-well PCR plate was loaded with 5 µl 2X Reaction Mix, 0.2 µl Superscript III RT/Platinum Taq Mix (with RNaseOUT Ribonuclease Inhibitor) (Invitrogen Cells Direct One-Step qRT-PCR kit, Life Technologies), 2.5 µl primer mix (containing 200 nM of each gene-specific primer), 1.3 µl $H_2O$. Single-cell suspensions were stained with 100 ng/ml DAPI. DAPI-negative cells were sorted into separate wells of the 96-well PCR plate. 32 cells were sorted into one well, to be used for serial dilution for generation of qRT-PCR standard curves. RNA reverse transcription and 22 cycles of cDNA pre-amplification were performed as previously described (*Dalerba et al., 2011*). The cDNA was diluted 1:5 in $H_2O$ and 2 µl were loaded into each well of a Lightcycler 480 384 Multiwell plate, with 4 µl Lightcycler 480 Probes Master, 0.08 µl Universal ProbeLibrary probe (Roche, Switzerland), 450 nM of each primer and $H_2O$ to 8 µl. qRT-PCR reactions were performed using a Lightcycler 480 II instrument with the following cycling conditions: 50°C for 2 min, 95°C for 10 min, then 40 cycles of 95°C for 15 s, 60°C for 1 minute.1:2 serial dilutions of the 32 cells sorted into one well were used for the generation of standard curves. Samples with *TBP* expression lower than 10% of the population average were removed from further analyses. Spearman's rank correlation coefficients for *TBP*-normalised gene expression values were calculated in R (http://www.r-project.org/) using the rcorr () function in the Hmisc package.

## Immunofluorescence, FACS, in situ hybridisation

For immunofluorescence analysis, cells were fixed in 4% paraformaldehyde (PFA) and incubated for 30 min in blocking buffer (PBS, 3% Donkey serum and 0.1% Triton). Primary antibodies were diluted in blocking buffer and applied for 1 hr at room temperature or overnight at 4°C. After three washes in PBS, secondary antibodies conjugated to Alexa fluorophores (Life Technologies) were diluted at 1:1000 in blocking buffer and applied for 1 hr at room temperature. The cells were washed at least three times in PBS and visualised on an Olympus inverted fluorescence microscope. For nuclear counter staining, cells were incubated in 1 µg/ml DAPI (Sigma) for 10 min after immunostaining. Embryos were fixed in 4% PFA for 2 hr at room temperature. The PFA was quenched with $NH_4Cl$ for 30 min at room temperature. For wholemount staining, the embryos were permeabilised in PBS+0.5% Triton for 1 hr at room temperature, then transferred to blocking buffer overnight at 4°C. They were

then incubated in primary antibodies for 48 hr at 4°C, subjected to five 10-min washes in PBS+0.1% Triton, incubated in secondary antibodies and 100 ng/μl DAPI for 48 hr at 4°C, then subjected to five 10-min washes in PBS+0.1% Triton. Staining of sectioned embryos (7 μm sections) followed a similar protocol: the slides with gelatine-embedded sections were incubated at 55°C for 15 min to melt the gelatine, then permeabilised with PBS+0.5% Triton for 10 min at room temperature. The blocking and staining procedure is the same as that described for cells. After staining, the sections were mounted in Prolong Gold Antifade Reagent (Life Technologies).

Fluorescent mages were taken using a monochrome Retiga2000 camera using an Olympus IX51 inverted microscope or a Leica SPE confocal microscope. Images from the Olympus microscope were collected using Volocity software. 3D rendering was done using the 3DViewer plugin in Fiji ImageJ. Fluorochromes were Alexa 488, 568, 594 or 647.

Primary antibodies were obtained from the following sources: Neuronal beta-III tubulin (TUJ1: Covance, Princeton, NJ), E-Cadherin (ECCD1 for blocking experiments, ECCD2 for immunofluorescence: BD Biosciences, Franklin Lakes, NJ), N-Cadherin (C32: BD Biosciences), Oct4 (Santa Cruz, Dallas, TX), T (R&D, Minneapolis, MN), Nanog (MLC-51: eBioscience, San Diego, CA), Esrrb (PP-H6705-00: Persaeus Proteomics, Japan), Klf4 (R&D), Flag (clone M2 HRP conjugated for Western Blot, or unconjugated for immunofluorescence: Sigma, St Louis, MO). Antibodies against Vimentin and Keratin8 were purchased from the Developmental Studies Hybridoma Bank, developed under the auspices of the NICHD and maintained by The University of Iowa, Department of Biological Sciences, Iowa City, IA 52242.

FACS analysis was performed using a Becton Dickinson FACS Calibur flow cytometer. FACS sorting was carried out on a FACS Aria.

In situ hybridisation for *Id1* was performed using a probe described in *Gray et al. (2004)*. The probe was generated using the following primers: F:AAGGTCGCGAGTGGCAGTGC R:GCCTGAAAAGTAAGGAAGGG.

## Quantification of immunostaining

In order to quantify the fluorescence signal in each individual cell, we generated an automated pipeline for image analysis as described previously (*Davies et al., 2013*) or used tools provided through the Farsight project (http://www.farsight-toolkit.org).

In order to quantify neurite density after neural differentiation, five randomly selected fields (for each condition: treatment and day of differentiation) stained for Tuj1 were pictured. Each field was scored depending on the number of neurites present from 0 to 9, where 0 means no neurites can be observed and 9 neurites cover almost the entire field; the scores for each condition were averaged.

## Acknowledgements

We thank Robert Benezra and Hyung-song Nam for the Id1-Venus targeting construct, Val Wilson for help with embryo dissection, Rolf Kemler for *Cdh1* null ES cells, Nicola Festuccia, Ian Chambers, Anna Waterhouse, Andrew Smith and Owen Davies for providing reagents and advice, Aliaksandra Radzisheuskaya for help with qRT-PCR experiments and Kumiko Iwabuchi for advice on single-cell qRT-PCR. We thank Josh Brickman, Tilo Kunath, and Val Wilson for helpful discussions and comments on the manuscript, and Alfonso Martinez Arias and Tristan Rodriguez for constructive review of this manuscript. SL is a Wellcome Trust Career Development Fellow (WT082232AIA) and GB is a Sir Henry Wellcome Postdoctoral Fellow (WT100133).

## Additional information

### Funding

| Funder | Grant reference number | Author |
|---|---|---|
| Wellcome Trust | WT082232AIA | Sally Lowell |
| Biotechnology and Biological Sciences Research Council | BB/I006680/1 | Sally Lowell |
| Wellcome Trust | WT100133 | Guillaume Blin |

The funders had no role in study design, data collection and interpretation, or the decision to submit the work for publication.

## Author contributions

MM, SL, Conception and design, Acquisition of data, Analysis and interpretation of data, Drafting or revising the article; PAN, GB, Conception and design, Acquisition of data, Analysis and interpretation of data; AP, XZ, Acquisition of data, Analysis and interpretation of data

## Additional files

### Supplementary files

• Supplementary file 1. Description of primers and Roche UPL probes used for qRT-PCR.

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
