## [Decision Letter]

Thank you for sending your work entitled “BMP suppresses differentiation of pluripotent cells by blocking an epithelial to mesenchymal transition” for consideration at *eLife*. Your article has been favorably evaluated by a Senior editor and 3 reviewers, one of whom is a member of our Board of Reviewing Editors.

The following individuals responsible for the peer review of your submission have agreed to reveal their identity: Freda Miller, Reviewing editor; Tristan Rodriguez and Alfonso Martinez Arias, peer reviewers.

The Reviewing editor and the other reviewers discussed their comments before we reached this decision, and the Reviewing editor has assembled the following comments to help you prepare a revised submission.

This manuscript addresses the mechanisms whereby BMP4 and its downstream target Id1 regulate the transition between pluripotent cells and their differentiated neural and mesodermal progeny. The study is largely pursued in murine ES cells, and provides evidence for the interesting conclusion that BMP4 and Id1 play an important role in regulating pluripotency and these early stages of differentiation via Cdh1 and potentially EMT.

1) A major limitation of this work as currently presented is that it is difficult to be definitive about EMT transitions in ES cells. For example, the repression of EMT might be an indirect consequence of the role of BMPs in promoting a posterior epiblast identity. One way the authors could address this question is to test if BMPs can repress EMT in the directed mesoderm assays shown in Figure 2. However, while this experiment might strengthen their general conclusion, we also feel that it would be better to rewrite the paper so that the conclusions regarding EMT (particularly with regard to neural induction) are more cautious and not couched as the major conclusion of the manuscript (such as in the title).

2) Much of this manuscript relies upon qPCR-based approaches for markers associated with certain states. However, although the authors have error bars on these and their other experiments, statistical significance is largely absent from the manuscript. Since their conclusions are largely quantitative ones, the authors need to explicitly indicate how many times each experiment was done, and to perform appropriate tests of significance.

3) In their studies on Id1, the authors establish a very nice control for their manipulated ES cells by floxing their Id1 transgene so that it can be deleted. They then show that this works with regard to neural differentiation in Figure 3—figure supplement 1. However, they never take advantage of this control for any of their EMT/mesoderm experiments. It would significantly strengthen the paper if they included these data, since in a sense this is the ultimate control.

4) The studies in Figure 4 looking at Oct4 versus Cdh1 during neural induction/neural tube establishment should be improved. Also, previous work from Derek van der Kooy's lab has shown that Cdh1 is expressed in the neural tube at later timepoints (Karpowicz et al., J. Neurosci., 2007). Do the authors also find this?

5) As a general comment, many of the figure legends are unclear. For example, in Figure 4, this appears to be the levels of expression of the various genes that are indicated in individual cells taken at 3 days that are sorted out for their relative levels of these markers. Is that the case? And how did the authors obtain the R values for correlation with Cdh1 as shown in 4B? Figure 5 cold also benefit from better labeling. Thus, in general, clarification in the figure legends would significantly improve “readability”.

---

## [Author Response]

*1) A major limitation of this work as currently presented is that it is difficult to be definitive about EMT transitions in ES cells*.

This is a very good point. Upon reflection, we realise that the terms “EMT” or “EMT-like” are not appropriate for the events that we are discussing: what we are really talking about are changes in E-Cadherin (Cdh1). We have now removed all inappropriate references to EMT throughout the text and in the title and replaced them with reference to changes in Cdh1. We only use the term EMT where we are referring specifically to conventional EMT events such as those that accompany formation of the mesoderm at the primitive streak, or where we refer to markers of EMT.

It is interesting, however, to note that the changes in Cdh1 that accompany neural specification are accompanied by some other changes that characterise conventional EMT, such as upregulation of N-Cadherin and Vimentin. We have included a brief discussion on this point at the start of the Discussion, whilst making it clear that these changes do not bear all the hallmarks of a classical EMT.

We appreciate this constructive advice and feel that our paper is now clearer and more to the point framed in terms of E-Cadherin rather than EMT.

*For example, the repression of EMT might be an indirect consequence of the role of BMPs in promoting a posterior epiblast identity*.

This is a perceptive point. One piece of evidence that argues against this interpretation is that suppression of Cdh1 alone is sufficient to rescue neural differentiation at moderate doses of BMP4 (Figure 6). If the primary effect of BMP is to promote posterior identity, with repression of EMT being a secondary consequence of this, then experimental suppression of Cdh1 in the presence of BMP might be expected to always drive mesoderm rather than neural differentiation. Since this does not happen (Figure 6) we tentatively conclude that repression of “EMT” is not entirely an indirect consequence of the role of BMPs in promoting a posterior epiblast identity. However, we fully accept the wider point that we should be more cautious in our conclusions regarding EMT. We have rewritten our manuscript accordingly as described above.

*One way the authors could address this question is to test if BMPs can repress EMT in the directed mesoderm assays shown in*
Figure 2*. However, while this experiment might strengthen their general conclusion, we also feel that it would be better to rewrite the paper so that the conclusions regarding EMT (particularly with regard to neural induction) are more cautious and not couched as the major conclusion of the manuscript (such as in the title)*.

This is a great suggestion. We added BMP to the directed mesoderm protocol based on Collagen IV and FCS (as in Figure 2). Exogenous BMP had profound effects during the protocol although unfortunately these effects appear to be somewhat toxic for the cells and so the data is difficult to interpret. We do not feel that these data are sufficiently informative to include in the paper, but we outline our findings below for your consideration.

Briefly, addition of BMP induced a clear morphological change in the cells. Rather than adopting the usual mesenchymal morphology (Figure 7), the cells became very large and flat (Figure 7). Whilst it is tempting to interpret this as an inhibition of EMT, these flatted cells do not have the typical morphology of healthy epithelial cells: rather they seem unhealthy, unable to proliferate, and destined to die. This is the type of morphology that we typically see in ES cell cultures that have been subjected to environmental or genetic stress.Author response image 1.

If BMP were simply inhibiting EMT during mesoderm differentiation we would expect these cells to remain, at least for some time, in a T+ pre-ingression epiblast-like state. However, qPCR analysis (for what it is worth from such unhealthy looking cells) reveals that BMP treatment promotes rapid loss of Oct4 (Figure R1C) and abolishes the upregulation of T-Brachyury that usually accompanies epiblast and subsequent mesoderm differentiation (Figure 7). We tested two different doses of BMP4 (10ng/ml and 50ng/ml) with similar results.

We conclude that BMP treatment in the context of this undefined directed mesoderm differentiation protocol strongly interferes with the normal differentiation process in ways that are difficult to interpret, but that clearly extend beyond a simple suppression of EMT. We think it most likely, based on morphological observations, that the cells adopt an unhealthy non-physiological state, perhaps due to exposure to conflicting environmental signals. We would therefore not feel comfortable drawing any conclusions from these experiments with relation to our proposed model.

More generally, we entirely agree with the reviewers' point that it is not appropriate for us to draw strong conclusions about control of EMT by BMP signalling, and we have followed their advice and rewritten the paper to remove reference to EMT as described above.

*2) Much of this manuscript relies upon qPCR-based approaches for markers associated with certain states. However, although the authors have error bars on these and their other experiments, statistical significance is largely absent from the manuscript. Since their conclusions are largely quantitative ones, the authors need to explicitly indicate how many times each experiment was done, and to perform appropriate tests of significance*.

We have included this information in the revised version of the manuscript. Significance is depicted on figures, and details are provided in figure legends and in the Materials and methods.

*3) In their studies on Id1, the authors establish a very nice control for their manipulated ES cells by floxing their Id1 transgene so that it can be deleted. They then show that this works with regard to neural differentiation in*
Figure 3*–figure supplement. 1. However, they never take advantage of this control for any of their EMT/mesoderm experiments. It would significantly strengthen the paper if they included these data, since in a sense this is the ultimate control*.

We agree that the control we used was not ideal. Whilst the floxed reverted line is a better control, this is also not the absolute ideal. We have recently developed a superior system that is much more tightly controllable. This is a dox-inducible Id1 overexpression system that allows us to compare the behaviour of the same cell line in the presence or absence of Id1 activation, described in Figure 3–figure supplement 1. We have used this new inducible system to generate a new set of data for Figure 3. We have additionally improved Figure 3 by including a full time course from d0 to d5 (our original data examined d0 to d3 only).

We are pleased to report that the data shown in the original version of Figure 3 is fully recapitulated with the better-controlled inducible system and indeed the effects are even more striking with the new system (see new Figure 3). This is despite the fact that this new inducible expression system provides more modest levels of exogenous Id1 than the original constitutive expression system (see new Figure 3—figure supplement 1). We therefore feel that the new data is not only better controlled but also even more strongly supportive of our model than our original data set.

When it comes to analysing longer-term consequences of Id1 overexpression on lineage bias we encounter a slight caveat with the dox inducible system. A significant proportion (around 50%) of cells lack expression of the flag-tagged transgene on day 5 of differentiation and the flag-negative population are able to generate neurons (Figure 3—figure supplement 1). Furthermore, the differentiation delay is more pronounced with the inducible system than with the constitutive system (Figure 3). Taken together, these two things make it difficult to use the dox-inducible system to test the eventual lineage preference of overexpressing cells. To strengthen this particular point we therefore turned back to the constitutive overexpression system, which a) allows for more uniform maintenance of Id1 activity throughout the culture (Figure 3—figure supplement 1) and b) allows for a more rapid ‘escape’ into differentiation (Figure 3—figure supplement 2). We performed a new series of experiments this time using the appropriate floxed-reverted control cell line as suggested by the reviewers.

These new better-controlled data confirm that cells that are able to persistently maintain exogenous Id1 activity eventually favour mesoderm rather than ectoderm or endoderm fates, in keeping with our original observations. These new data are provided in the new version of Figure 3—figure supplement 2.

*4) The studies in*
Figure 4
*looking at Oct4 versus Cdh1 during neural induction/neural tube establishment should be improved*.

We agree that the correlation between Oct4 and Cdh1 during neural development was difficult to see clearly in the original version of Figure 4. We have now strengthened these data by repeating these experiments and performing quantitative analysis. We first quantified the expression of Oct4 and Cdh1 in the epiblast at E7.5 to provide a baseline starting figure. We then compared this with expression of Oct4 and Cdh1 within the developing neural tube at three developmental stages spaced approximately four hours apart, spanning the time period over which Oct4 is lost from the developing neural tube, i.e., during formation of the first 12 somites. Note that our timing is assessed by somite number, and that the first several somites form in the mouse at hourly intervals (Tam 1981).

All antibody staining was performed simultaneously to minimize artifactual variation between time points, but we also directly controlled for technical variability between sections by quantifying E-Cadherin expression in the surface ectoderm immediately adjacent to the neural tube and confirming that these measurements are similar across all time points.

These new data confirm a striking correlation between the decline of Oct4 and the decline of Cdh1 within the neural tube.

*Also, previous work from Derek van der Kooy's lab has shown that Cdh1 is expressed in the neural tube at later timepoints (Karpowicz et al., J. Neurosci., 2007). Do the authors also find this*?

It is indeed interesting that E-Cadherin becomes re-expressed in foetal and adult neural stem cells as reported by Karpowicz et al. We have not examined these later stages of foetal and adult development ourselves as our focus is on the transition from pluripotency to lineage commitment. Interestingly, these authors also report a steep decline in E-Cadherin expression in the early neural tube by E9.0 (see Figure 2 in [18]). This fit exactly with our own observations (Figure 4) and we apologise for failing to reference this work in our original manuscript:Karpowiczet al.are cited in our revised manuscript and we modify our text to clarify that “Cdh1 is downregulated during the early stages of neural tube formation”.

*5) As a general comment, many of the figure legends are unclear. For example, in*
Figure 4*, this appears to be the levels of expression of the various genes that are indicated in individual cells taken at 3 days that are sorted out for their relative levels of these markers. Is that the case? And how did the authors obtain the R values for correlation with Cdh1 as shown in 4B?*
Figure 5
*cold also benefit from better labeling. Thus, in general, clarification in the figure legends would significantly improve “readability”*.

We have improved the figure legends and hope that they are now easier to follow.